# Characterization of Fish Spawning Grounds near the Likouala-Aux-Herbes River, Lac Tele Community Reserve (LTCR), Republic of Congo, for Sustainable Wetland Management

**Eric Bertin Ndzana Biloa [1,*], Victor Mamonekene [2] and Jean-Claude Micha [3]**

[1] Regional Post-University School of Integrated Management of Tropical Forests and Lands (ERAIFT), Campus UNIKIN BP., Kinshasa 15 373, Democratic Republic of the Congo

[2] National High School of Agronomy and Forectry, University Marien Ngouabi, Brazzaville BP 69, Congo; vito.mamonekene@gmail.com

[3] Faculty of Sciences, Department of Biology, University of Namur, (UNamur), B-5000 Namur, Belgium; jean-claude.micha@unamur.be

[*] Correspondence: ericbertinndzanabiloa@gmail.com

**Abstract:** A land use analysis of Lac Télé Community Reserve (LTCR) using the Landsat Thematic Mapper (TM), ETM+ images (Enhance Thematic Mapper), and OLI images highlighted significant changes in plant cover between 1986, 1999, and 2019. The rate of forest area increased by 5% from 1986 to 1999 but decreased by 3% from 1999 to 2019 for the entire LTCR, benefitting the growth of savannahs, which respectively increased by 5% between 1986 to 1999 and 12% between 1999 to 2019. The conversion of this forest area to savannah due to the practice of slash and burn agriculture can be justified by the opening of this forest area, but it contributes greatly to the degradation of fish species spawning grounds in the Likouala-aux-Herbes River. According to characterization of the 151 spawning grounds identified, the physical and chemical water parameters, which have been defined as temperature (28.13 °C), hydrogen potential (4.23), and depth (3.34), did not change significantly in the study villages between July and September 2019. The unregistered ichthyological diversity observed during the study in the seven pilot villages, is due to the diversity of the microhabitats noted in the villages of the LTCR, especially in the villages of Botongo, Mossengue, and Bouanela, where the indices of ichthyological diversity were the highest.

**Keywords:** remote sensing; geographic information systems (GIS); fish biodiversity





## 1. Introduction

The Lac Télé Community Reserve (LTCR), with coordinates 01°11′ N 17°13′ E, is a category VI protected area, according to the IUCN classification, and is primarily managed by the Congolese State through the Congolese Agency for Wildlife and Protected Areas (ACFAP). Thanks to the LTCR Support Project, ongoing since 2001, it also benefits from the support of the American NGO Wildlife Conservation Society (WCS) in the management of the protected area [1]. The LTCR comprises an area of 438,960 ha [2]. It is located in the department of Likouala, whose capital is Impfondo, and which has two administrative subdivisions, categorized as districts: Epéna, located 85 km from Impfondo, and Bouanela [3]. The vegetation of the LTCR is part of the natural Congolese region called Lower Sangha, consisting of the Cameroonian-Congolese sector and the Guineo-Congolian region [4]. The reserve contains a wide variety of vegetation habitats, including swamp floodplain forests, which make up 49% of the area; the forests, susceptible to flooding during the rainy season, which occupy 17% of the reserve and which are made up mostly of gallery or riparian forests located along the watercourses of the LTCR [5]; the mixed forests on dry land, which occupy about 10% of the reserve; and finally, the savannahs, which represent about 16% of

the area [6]. This diversity of habitats plays a very large role in the abundance, distribution, and diversity of fauna in general [7] and fishery resources in particular, which constitute the bulk of the animal protein supply near the LTCR. However, similar to trends of fishing catches at the global level, those recorded within the LTCR are also decreasing [8]. What could be the cause of the degradation of fishery resources within the LTCR, particularly in the Likouala-aux-Herbes River [9,10]? Is it the degradation of the environment over time, including the effects of climate change, around the watershed of the Likouala-aux-Herbes River that is causing the decline in catches? Is it the overexploitation of fish stocks resulting from an uncontrolled intensification of catches? Or is it the reduction and exploitation of spawning grounds, necessitating the use of restoration and recruitment within the fisheries of the Likouala-aux-Herbes River? The overall objective of this study is to contribute to the sustainable management of fisheries resources within the LTCR. More specifically, through an analysis of the land use analysis in the watershed of the Likouala-aux-Herbes River, we conducted a mapping of the spawning grounds to ensure that they are protected for the future through the use of adequate conservation practices.

## 2. Materials and Methods

According to the environmental context of the LTCR (Figure 1), including the distances between the seven pilot villages, 21 sampling stations were established in the study villages, at a rate of 3 stations per village [11].

- 1st station: upstream of the village (northern limit of the spawning grounds) (i).
- 2nd station: village area (near the main port of the village) (ii).
- 3rd station: downstream of the village (southern limit, of the h spawning grounds) (iii).

Using the village as the sampling unit, the three values recorded in each locality were cumulated into an average, which served as a reference value for the physicochemical characterization (water height (m), width (m), speed (m/s), flow ($m^3$/s), transparency (m), color, temperature (°C), pH, conductivity at 25 °C, nature of the substrate, and the occupation of the ground of the station) of the waters of each pilot village. These data were recorded monthly throughout the study period, according to the recommendations of Zebaze [12], who concluded that this frequency accounts for the functioning of the hydrosystems. Data for the physicochemical parameters, namely temperature, pH, and conductivity at 25 °C, were measured using the HACH IntelliCAL phC101 multi-parameter (New Delhi—India) (b). Water depth (c) and transparency (d) were measured using a homemade water level gauge and a Secchi disk, respectively. The width of the sampled wetted section was measured using a tape measure. Using a stopwatch, the water speed was measured in m/s, employing an empirical method which consisted of recording the distance and the time traveled by an immersed float device that was followed over a certain distance. Subsequently the flow rate was calculated using the following formula [13].

$$Q = V. S \qquad \text{with} \qquad V = \text{Flow velocity in m/s}$$

$$S = \text{wetted area in m}^2$$

$$Q = \text{flow in m}^3/\text{sec}$$

The fish samples were collected from July 2019 to September 2019 using several different techniques, including the use of gillnets measuring 20 m long and 1 m high, with meshes between knots measuring 1.5 cm, 2 cm, 2.5 cm, 3 cm, and 6 cm; traditional traps about 1 m long and 30 cm in diameter; and individual lines and longlines measuring 25 m long, with number 8, 10, 12, 14, 16, 18, and 20 hooks, which were baited with earthworms, mollusks, and small fish. Harpoons were also used to capture large specimens [14–19].

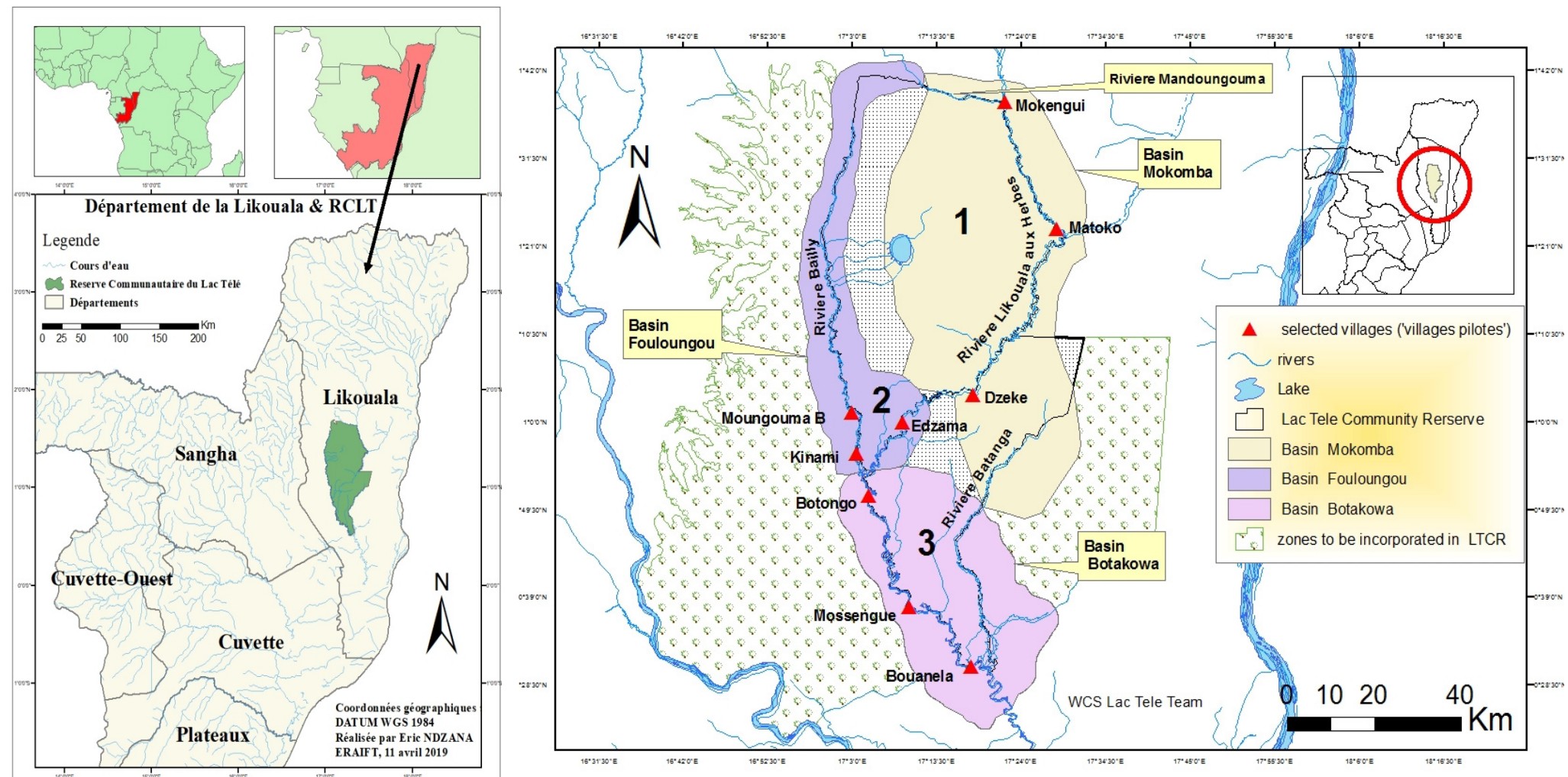

**Figure 1.** Location of study area in the Republic of the Congo.

The methodological approach adopted in this study was based on a cartographic, analytical, and diachronic approach focused on the use of remote sensing and GIS. Three Landsat MSS satellite images from 1986, 1999, and OLI 2019 were utilized. The resolution of each image was 30 m. The projection adopted was WGS 1884, UTM-Zone 33 N. ArcGIS 10.5 software was used for the extraction, digitalization, combination, and integration of the different shapefiles. The land cover maps were developed using satellite images. The land use class areas were calculated using ArcGIS 10.5 [20]. This allowed for the extraction of the diachronic results of the landscape analysis from different years near the watershed of the Likouala-aux-Herbes River. Over 33 years, two periods are identified: 1986–1999, and 1999–2019. As shown in map 1, two Landsat scenes made it possible to acquire images corresponding to the geographical area of the Lac Télé Community Reserve, namely: Scene 1 (Path: 181 raw: 59) and Scene 2 (Path: 181 raw: 60). The semiautomatic classification was completed using ArcGIS Analysis and Classify to produce the polygons to compare the landcover at different times.

- Changes: the mode of space occupation remained the same between the two years (initial state).
- Modification: the mode of occupation of space has changed from one class to another, but remained in the same category.
- Conversion: the mode of occupation of the space of a class has converted to another class in a different category (example: Gallery forest became herbaceous or tree savannah). To quantify land use changes, the annual expansion rate method was used. It consisted of calculating the annual expansion rate using the formula proposed by the FAO in 1996, as follows: T = S2 − S1; where T = rate, S1 = area of class in the first year, and S2 = area of class in the second year. Regarding the value of S (S in unit), if S is a positive value, there was an increase in the area of the classes during the period analyzed, and if S is a negative value, there was a loss of area in the classes between the two periods. A value close to zero means that no change in the classes was noted between the two periods [21].

## 3. Results

### 3.1. Descption of Differents Results

3.1.1. Spatiotemporal Dynamics of Land Use in the Likouala-aux-Herbes Watershed between 1986, 1999, and 2019

- The distribution of land use (Figure 2) classes between 1986, 1999, and 2019 shows a relative conversion of forest (dense, swampy, floodable, and gallery) of the LTCR environment into savannah (herbaceous and shrubby).
- We observe a predominance of dense and swampy forests and a savannah dynamic around the watershed of the Likouala-aux-Herbes River.
- According to the analysis shown in 1986 (Table 1), the forest class was divided into dense forest, covering 53.4% (241,021.98 ha) of the LTCR; swamp forest, comprising 24% (108,334.26 ha) of the LTCR; and floodable forest, covering 2.5% (11,442.78 ha) of the area. Of the total (79.5%), in 1999, this progressed to 85%, comprised of 39.02% dense forest (176,187.78 ha), 34% swamp forest (153,325.44 ha), and 12.14% floodable forest (54,846.81 ha).
- During the same period, the shrub and herbaceous savannah classes increased from 0.018% in 1986 to 5% of the area of the LTCR in 1999 (Table 2), from 642.06 ha to 25,403.04 ha, respectively, and increasing to 17.8% in 2019, with 9.4% of shrub savannah (42,444.9 ha) and 8.4% (38,128.77 ha) of herbaceous savannah.
- The wetlands class occupied 20% of the LTCR in 1986 (90,012.06 ha), 10% in 1999 (41,690.07 ha), and 1% (4701.42 ha) in 2019.

3.1.2. Characterization of the Ichthyological Diversity of the Likouala-Aux-Herbes River

- A total of 1106 fish were harvested and divided into 25 genera belonging to 13 families (Tables 3 and 4);

- The Clariidae family was the most abundant, with 381 specimens, representing 34.07%, followed by the Mochokidae family, with 208 individuals, or 19.67%. The Bouanela and Mossengue stations contain the largest numbers of individuals, representing respectively 20.61% and 31.1%, while the Moungouma-Bailly and Dzeke stations recorded the lowest numbers of 8 and 28 individuals, i.e., 0.72% and 2.53% (Figure 3 and Tables 4 and 5).
- In axis II of the principal component analysis (Component 2, Figure 3), the distribution of the Alestidae and Mormyridae families was influenced by the presence of sandy type substrates, foliage, and macrophytes, coupled with the increase in temperature of the water (Tables 4 and 5).
- The presence of Alestidae (Component 2, Figure 3) is correlated with temperature peaks, namely 28.79 ± 1.7 °C for the Botongo station and 28.13 ± 1.17 °C for the Epena station (Table 5). As for the Mormyridae, the highest diversity indices in this case, 0.6229 for Mossengue and 0.5993 for Botongo, were recorded at the stations with the highest temperature peaks.
- The results of the principal component analysis (PCA) regarding the spatial distribution of the captured species made it possible to select the environmental and biotic variables which best express the links that exist between species and the parameters of the various aquatic habitats.
- Overall, considering all the stations, the environmental variables which influenced the distribution of the numbers of fish species recorded during the study period are the temperature, the width of the wetted section, the depth, the flow rate, and the composition of the different substrates encountered in the stations.
- reveals that the 13 families of fish species recorded in the different pilot villages were distributed according to a combination of the physicochemical parameters and nature of the substrate. Thus, following axis I (Component 1, Figure 4), the Cichlidae, Channidae, Claroteidae, and Mochokidae samples presented more affinity with the width of the wetted section of the medium, the depth, the flow, and the substrate of a clay–muddy nature.
- The highest specific richness values, namely (4 taxa) for Cichlidae and (3 taxa) for Mochokidae, were recorded in the Botongo and Mossengue stations, whose substrate composition is mud–sandy and exhibits the highest average widths.
- In axis II of the PCA (Component 2), the distribution of the Alestidae and Mormyridae families seems to be influenced by the presence of sand, foliage, and macrophyte type substrates, coupled with the increase in temperature of the water. The presence of Alestidae, is correlated with temperature peaks, namely 28.79 ± 1.7 °C for the Botongo station and 28.13 ± 1.17 °C for the Epena station. As for the Mormyridae samples, the highest diversity indices, 0.6229 for Mossengue and 0.5993 for Botongo, were recorded in the stations whose temperature peaks are among the highest.

The families of Distichodontidae, Hepsetidae, Notopteridae, Anabantidae, Schilbeidae, and Polypteridae show no affinity either with the different substrates recorded or with the physicochemical parameters of the water in axis III, in contrast to the distribution of the previous families presented in axes I and II. However, they have affinities with each other; hence, their grouping as an agglomerate in axis III.

### 3.2. Figures, Tables, and Schemes

The spawning ground where fish reproduce, feed, and ensure the development of their fry. These sites are generally subservient to the natural environment in which they exist and also serve as shelter during predator attacks. They are generally found along the shore, on the sides of the banks of watercourses, in floodplains, and in substrates containing vegetation, rocks, sand, or benthos (Figure 5 and Table 5). The spawning grounds identified in the LTCR (Figure 5) between July and September 2019 are mainly present in the gallery forests that run along three sub-basins of the Likouala-aux-Herbes River. In agreement with the fishing communities of the LTCR, as part of the project for the development and validation of the LTCR management

plan, certain spawning grounds have been declared accessible, regardless of the season during the year, to allow continued fishing. However, in order to guarantee the sustainability of fishing activities within the LTCR, certain spawning grounds have also been set aside through a fishermen's charter, again in agreement with fishing communities, using a participatory decision process. Figure 5 shows the location of these two types of spawning grounds.

**Table 1.** Distribution of LTCR land cover classes between 1986 and 1999.

| | Translation | Areas (ha) | | Expansion Rate | |
|---|---|---|---|---|---|
| CLASSES | CLASSES | 1986 | 1999 | Hectares (ha) | Percentage (%) |
| Zones humides | Wetlands | 90,012.06 | 41,690.07 | −48,321.99 | 53.68390636 |
| Forêt dense | Dense forest | 241,021.98 | 176,187.78 | −64,834.2 | 26.89970433 |
| Forêt marécageuse | Swamp forest | 108,334.26 | 153,325.44 | 44,991.18 | 41.52996476 |
| Forêt inondable | Floodable forest | 11,442.78 | 54,846.81 | 43,404.03 | 379.3136808 |
| Savane arbustive | Shrub savannah | 378 | 11,488.32 | 11,110.32 | 2,939.238095 |
| Savane herbacée | Herbaceous savannah | 264.06 | 13914.72 | 13,650.66 | 5169.529652 |

**Table 2.** Distribution of LTCR land cover classes between 1999 to 2019.

| | Translation | Areas (ha) | | Expansion Rate | |
|---|---|---|---|---|---|
| CLASSES | CLASSES | 1999 | 2019 | Hectares (ha) | Percentage (%) |
| Zones humides | Wetlands | 41,690.07 | 4,701.42 | -36,988.65 | 88.72292611 |
| Forêt dense | Dense forest | 176,187.78 | 119,643.84 | -56,543.94 | 32.0929976 |
| Forêt marécageuse | Swamp forest | 153,325.44 | 152,263.26 | -1,062.18 | 0.692761749 |
| Forêt inondable | Floodable forest | 54,846.81 | 94,270.95 | 39,424.14 | 71.88046123 |
| Savane arbustive | Shrub savannah | 11,488.32 | 42,444.9 | 30,956.58 | 269.4613312 |
| Savane herbacée | Herbaceous savannah | 13,914.72 | 38,128.77 | 24,214.05 | 174.0175153 |

( ) progress; (-) regression; (0,0): no major change.

**Table 3.** List of families and species of fish caught in the Likouala-aux-Herbes River near the seven pilot villages between July and September 2019.

*Polypteridae*
*Polypterus delhezi* (Boulenger, 1899)
*Mormyridae*
*Petrocephalus ballayi* (Sauvage, 1883)
*Petrocephalus microphthalmus* (Pellegrin, 1908)
*Mormyrops boulengeri* (Pellegrin, 1900)
*Marcusenius moorii* (Günther, 1867)
*Marcusenius* sp
*Gnatonemus petersii* (Günther, 1862)
*Hippopotamyrus weeksii* (Boulenger, 1902)
*Notopteridae*
*Papyrogranus afer* (Günther, 1868)
*Xenomystus nigri* (Günther, 1868)
*Hepsetidae*
*Hepsetus odoe* (Bloch, 1794)
*Alestidae*
*Alestes macrophthalmus* (Günther, 1867)
*Distichodontidae*
*Ichthyoborus ornatus* (Boulenger, 1899)
*Distichodus noboli* (Boulenger, 1899)
*Distichodus altus* (Boulenger, 1899)
*Xenocharax spilurus* (Günther, 1867)
*Anabantidae*
*Microctenopoma nanum,*(Guichenot, 1859)
*Ctenopoma kingsleyae* Günther, 1896

*Claroteidae*
*Chrysichthys* sp.
*Chrysichthys cranchii* (Burchell, 1822)
*Parauchenoglanis punctatus.* (Sauvage, 1879)
*Parailia occidentalis* (Pellegrin, 1901)
*Schilbe marmoratus* Boulenger, 1911)
*Schilbe grenfelli* (Boulenger, 1900)
*Clariidae*
*Clarias buthupogon* (Sauvage, 1879)
*Clarias* spp. (Sauvage, 1879)
*Mochokidae*
*Synodontis nigriventris* (David, 1936)
*Synodontis camelopardalis* (Poll, 1971)
*Synodontis falvitaeniatus* (Boulenger, 1919)
*Channidae*
*Parachanna obscura* (Günther, 1861)
*Cichlidae*
*Hemichromis bimaculatus* (Gill, 1863)
*Hemichromis elongatus* (Guichenot, 1859)
*Tilapia congica,* (Gill, 1863)
*Tylochromis lateralis* (Sauvage, 1879)

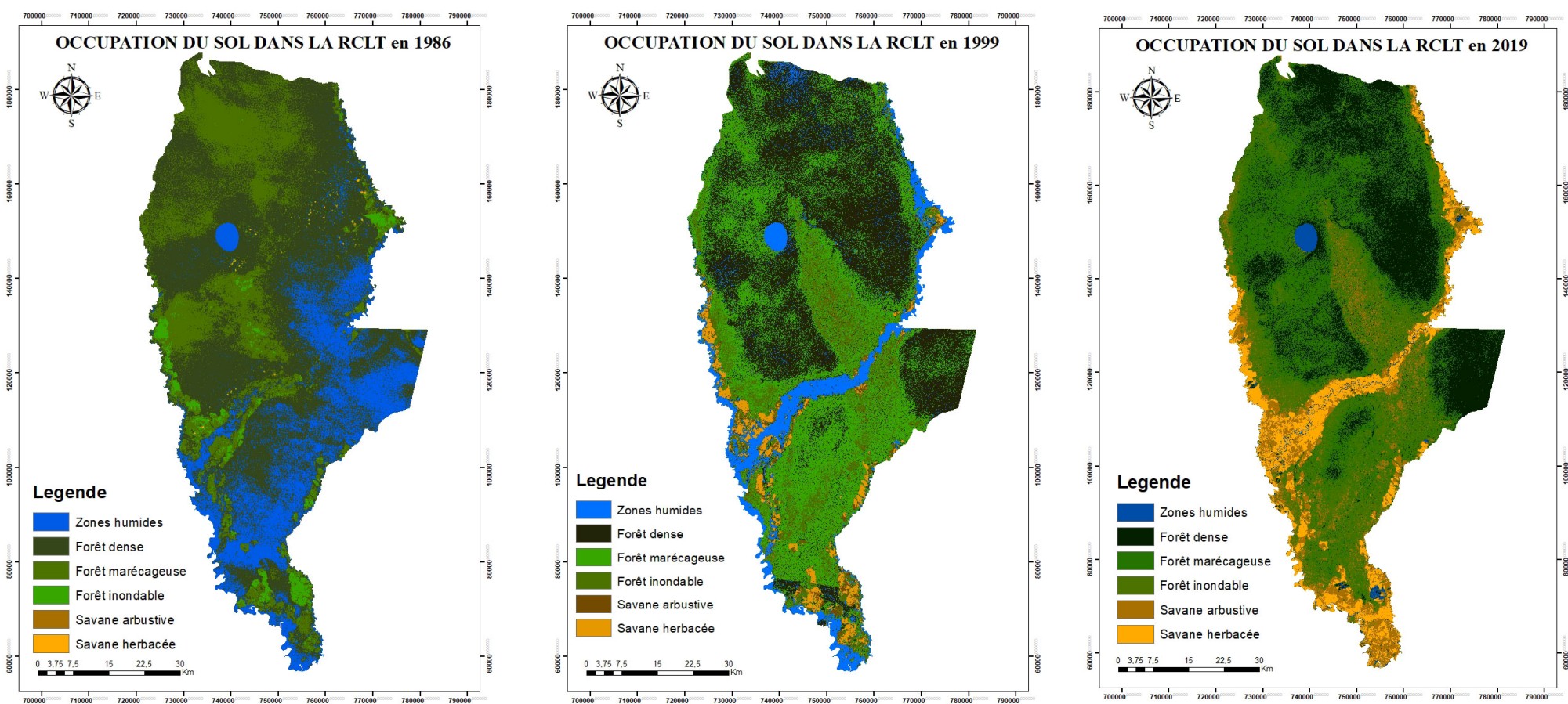

**Figure 2.** Land cover dynamics analysis of LTCR in "1986", "1999" and "2019".

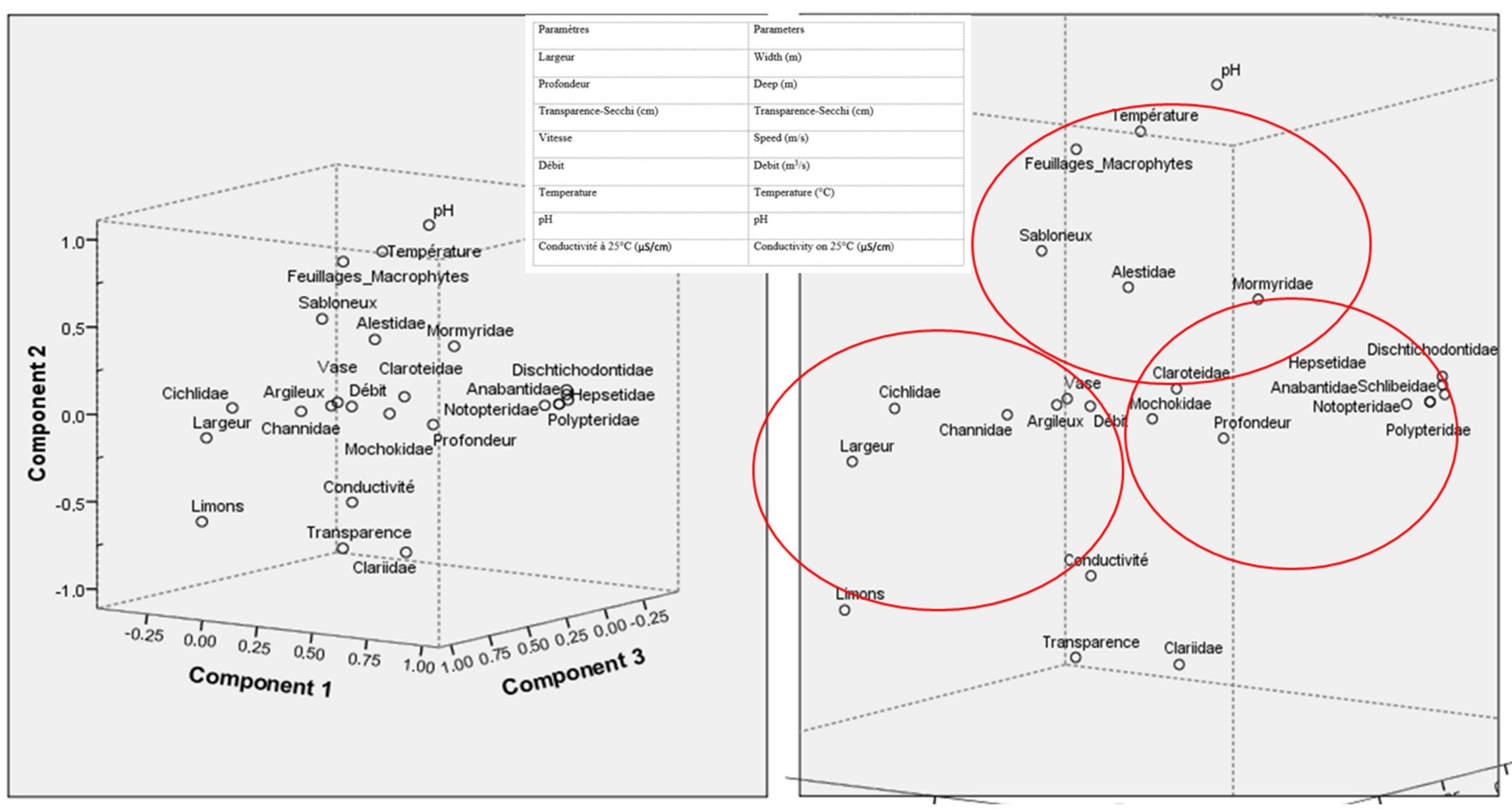

**Figure 3.** Analyses comparing hydromorphology and the distribution of fish families from the Likouala-aux-Herbes River recorded in the seven pilot villages of the LTCR from July to September "2019".

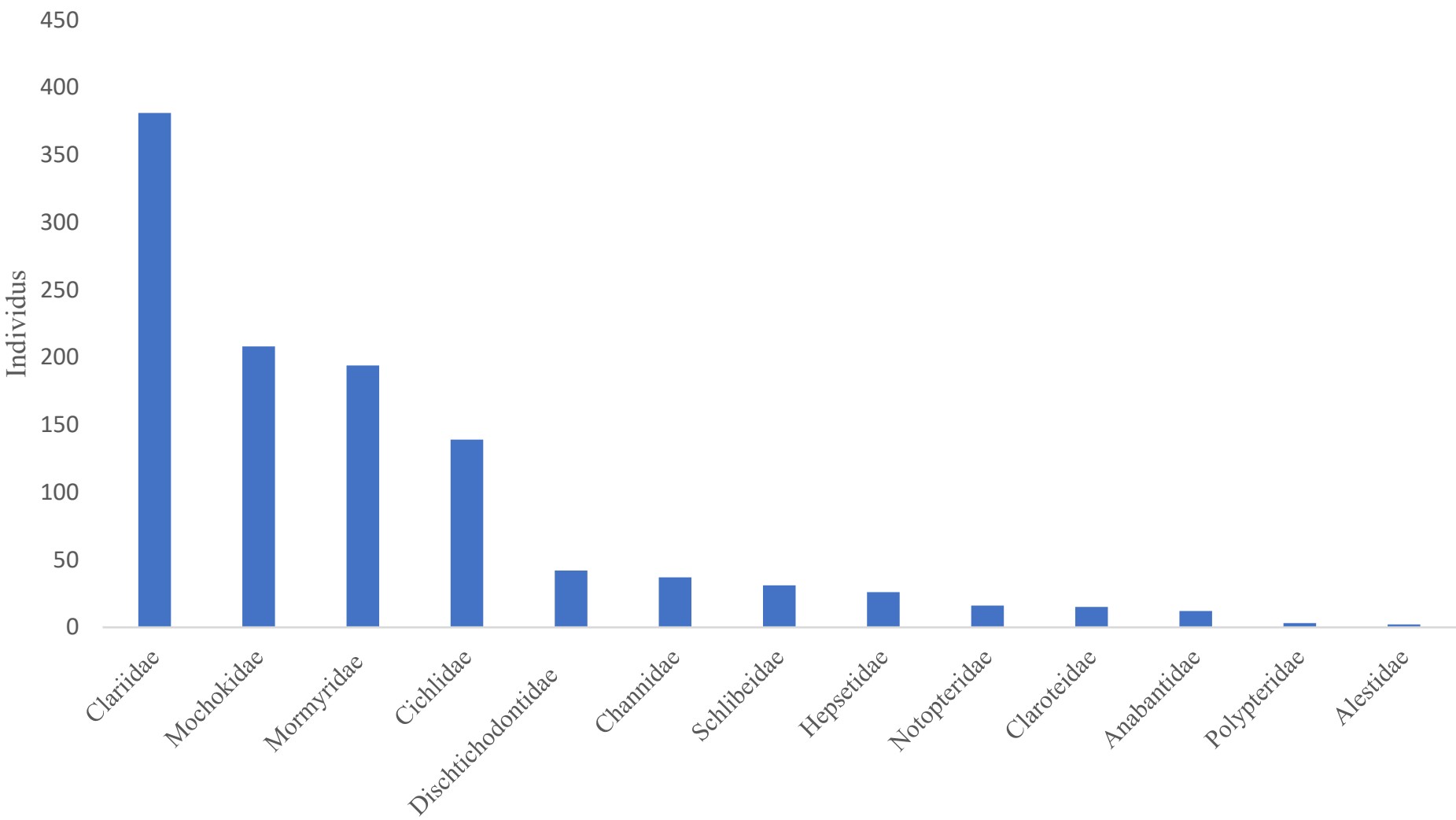

**Figure 4.** Fish families identified in the Likouala-aux-Herbes River near the seven pilot villages of the LTCR from July to September "2019".

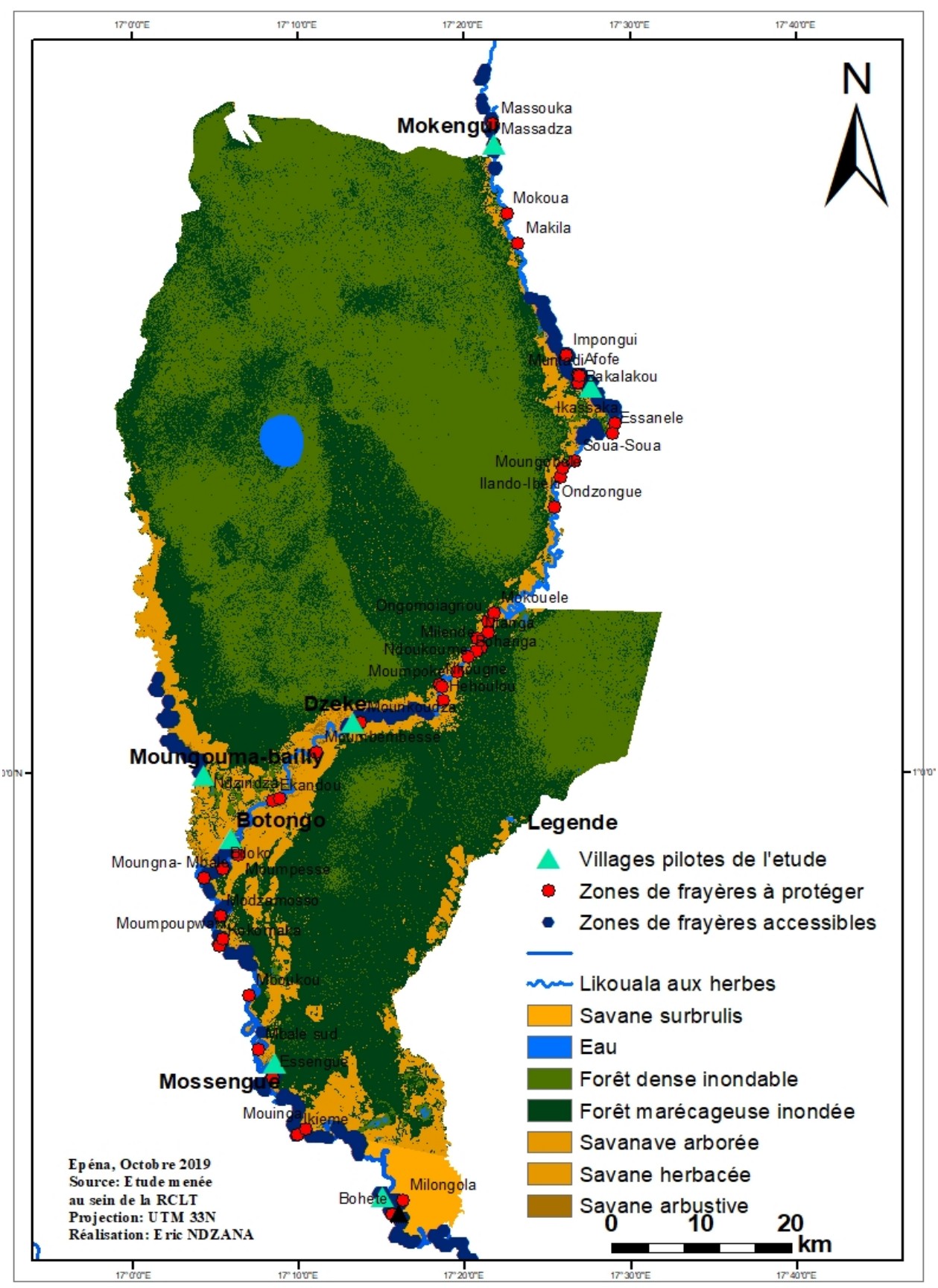

**Figure 5.** Mapping of prohibited and accessible spawning grounds in the Likouala-aux-Herbes River near the seven pilot villages of the three fish basins of the LTCR.

Table 4. Diversity of fish populations identified in seven stations of the Likouala-aux-Herbes River (LTCR) from July to September 2019.

| VILLAGES/FAMILIES | MOKENGUI | EPENA | DZEKE | MOUNGOUMA BAILLY | BOTONGO | MOSSENGUE | BOUANELA | TOTAL |
|---|---|---|---|---|---|---|---|---|
| Alestidae | | 1 | | | 1 | | | 2 |
| Anabantidae | | | | | | 12 | | 12 |
| Channidae | | | 1 | | 2 | 14 | 20 | 37 |
| Cichlidae | | | 6 | | 11 | 34 | 88 | 139 |
| Clariidae | 206 | 32 | 18 | 3 | 45 | 59 | 18 | 381 |
| Claroteidae | | | | | 13 | 1 | 1 | 15 |
| Dischtichodontidae | | 6 | | | 13 | 23 | | 42 |
| Hepsetidae | | | 3 | | 9 | 14 | | 26 |
| Mochokidae | | 3 | | | 22 | 100 | 83 | 208 |
| Mormyridae | 1 | 64 | | 5 | 55 | 51 | 18 | 194 |
| Notopteridae | | | | | 8 | 8 | | 16 |
| Polypteridae | | | | | | 3 | | 3 |
| Schlibeidae | | 2 | | | 4 | 25 | | 31 |
| TOTAL | 207 | 108 | 28 | 8 | 183 | 344 | 228 | 1106 |
| | 18.1 | 9.76 | 2.53 | 0.72 | 16.54 | 31.1 | 20.61 | 100.00 |
| *Indices of diversity (H′)* | | | | | | | | |
| Taxa (N) | 2 | 6 | 4 | 2 | 11 | 12 | 6 | |
| Individus (N) | 207 | 108 | 28 | 8 | 183 | 344 | 228 | |
| (H′) | 0.0132 | 0.455 | 0.4223 | 0.287 | 0.8477 | 0.9118 | 0.5964 | |

**Table 5.** Compared average hydromorphological and physicochemical characteristics of the spawning grounds of the Likouala-aux-Herbes Rivers near the seven pilot villages of the three LTCR fish basins from July to September 2019.

| Parameters/Stations | Mokengui | Epena | Dzeke | Moungouma-Bailly | Botongo | Mossengue | Bouanela |
|---|---|---|---|---|---|---|---|
| Width (m) | 42.32 ± 21.4 | 42.32 ± 19.23 | 42.30 ± 19.2 | 40.01 ± 21.28 | 43.15 ± 20.99 | 42.36 ± 21.55 | 45.36 ± 23.01 |
| Depth (m) | 3.43 ± 0.73 | 3.43 ± 0.73 | 3.43 ± 0.73 | 3.25 ± 0.75 | 3.26 ±0.73 | 3.24 ± 0.76 | 3.30 ± 0.78 |
| Transparency-Secchi (cm) | 88.70 ± 12.94 | 66.56 ± 8.50 | 66.51 ± 8.65 | 64.74 ± 10.15 | 65.69 ± 10.57 | 66.76 ± 11.63 | 68.48 ± 13.76 |
| Speed (m/s) | 0.23 ± 0.08 | 0.23 ± 0.08 | 0.23 ± 0.08 | 0.20 ± 0.08 | 0.2 ± 0.09 | 0.19 ± 0.1 | 0.16 ± 0.1 |
| Flow (m$^3$/s) | 24.04 ± 16.78 | 24.04 ±15.54 | 24.03 ± 15.52 | 17.81 ± 13.08 | 19 ± 13.56 | 17.98 ± 14.05 | 17.30 ± 14.52 |
| Temperature (°C) | 25.82 ± 2.61 | 28.13 ± 1.17 | 28.11 ± 1.88 | 28.86 ± 1.62 | 28.79 ± 1.7 | 28.64 ± 1.87 | 28.69 ± 2.21 |
| pH | 4.2 ± 0.42 | 4.53 ± 0.14 | 4.53 ± 0.19 | 4.48 ± 0.16 | 4.46 ± 0.16 | 4.42 ± 0.15 | 4.35 ± 0.13 |
| Conductivity at 25 °C (µS/cm) | 90.40 ± 27.65 | 68.16 ±23.45 | 68.12 ± 23.46 | 61.49 ± 22.58 | 57.65 ± 21.05 | 58.52 ± 21.75 | 58.20 ± 24.50 |
| Water color observed with the naked eye | Very high yellow (golden) | High yellow (golden) | Average yellow (bright) | Average yellow (bright) | Average yellow (bright) | Average yellow (almost clear) | Average yellow (almost clear) |
| Land cover classes | Gallery forest | Gallery forest, flooded grassy savannah | River (course bed and banks) | River, flooded grassy savannah, gallery forest | River, backwaters | River, flooded grassy savannah, backwaters | River, flooded grassy savannah, backwaters |
| Substrates of spawning grounds | Mud, sand, clay, mud and silt | Mud, sand, clay, sandy mud, foliage and macrophytes | Mud, sand, clay, sandy mud, foliage and macrophytes | Mud, sand, foliage and macrophytes mud | Mud, sand | Foliage and macrophytes mud, mud, sand, sandy silt | Sand, silt, Sand, silt and silt, foliage and macrophytes |
| Other information on the surrounding vegetation observed in the spawning area | Variety of forest species with a dominance of the species with local name "Môtôlô" | Variety of forest species and grassy formations with a dominance of grasses | Variety of partially burned forest formations with a dominance of helophytes | Variety of grassy formations with a dominance of helophytes Hydrophytes and submerged macrophytes | Variety of savannah formations with a dominance helophytes and floating hydrophytes (reeds) | Variety of partially burned savannah formations with a dominance Macrophytes Helophytes and floating hydrophytes (reeds) | Variety of partially burned savannah formations with a dominance macrophytes helophytes floating hydrophytes, (reeds) |
| Accessible spawning grounds | 11 | 28 | 16 | 17 | 24 | 28 | 27 |
| Sum of water areas of spawning grounds in (m$^2$) | 497,998.07 | 2,512,671.414 | 502,085.049 | 848,104.1616 | 1,559,800.586 | 1,094,650.936 | 977,408.9037 |
| Sum of water areas of spawning grounds in (ha) | 4.9 | 25.12 | 5.02 | 8.4 | 15.59 | 10.94 | 9.7 |

## 4. Discussion

The analysis of land cover from 1986 to 2019 showed that the LTCR remained mostly stable at 86% over time. However, in 33 years, the evaluation of the spatiotemporal dynamics of the vegetation of the LTCR has revealed the regression of gallery forests and the mosaics of seasonally flooded dense forests, which occupy almost 13% of the LTCR in favor of savannah formations (grassy, tree, shrub, and slash and burn). These results are similar to those obtained by Taibou Ba [22]) in the Saloum Delta, in which the conversion of open and dry forests to savanna formations between 1979 and 1999 is presented. This dynamic regarding land cover was also present in the center of Togo [23].

In 24 years, the area of open and dry forests has degraded and has decreased by 157,988 ha, representing a regression rate of 23.44%. The reasons for these changes to the environment are identical, regardless of the sites, and are likely due, in part, to climate change, but also, and above all, to the intense and permanent exploitation of natural resources by humans who systematically set fire to this area to collect fish in small permanent ponds that are in the process of drying up and are difficult to access. Badahoui [24] also described similar causes of the degradation of fishery stock in lake Ahémé in Benin.

The LTCR forests provide many ecosystem goods and services that benefit all people in and around the LTCR villages. In addition to non-timber forest products (NTFPs), forests are spiritual places for communities [25]. Many indigenous communities (pygmies) present in the villages of the LTCR exhibit lifestyles entirely dependent on the forest [26]. Certain forest species used for pharmacopoeia, wood-energy, and materials for various activities (construction, crafts, etc.) are regularly extracted in the gallery and dense flood-prone forests found in the LTCR. The exploitation conditions of forest areas, which sometimes become completely flooded, impose considerable impact on the fishing activities of the communities of the LTCR in the dry season, requiring a great deal of effort to develop them. In order to capitalize on the effort, they use bush fires to recover some fish and create agricultural areas [27]. The extensive slash-and-burn agriculture technique obliged people in the reserve to exploit several hectares each year. This is one of the causes of plant cover degradation and therefore, the soil's impoverishment. The increase in population density could also be a driving force of this degradation, which does not exclusively affect the terrestrial forest environment, but also the aquatic environment.

The fish resources of the LTCR, and the large terrestrial fauna, regulate their lifestyles according to the forest environment. The edge that comes from the decomposition of organic matter of dead trees or parts of trees (leaves, bark, branches, roots and fruits) contributes to strengthening the aquatic food chain on which biodiversity in general and ichthyology in particular strongly depend. By destroying gallery forests through agricultural practices based on the use of bush fires, certain species of fish, whose feeding and reproduction are on ligneous support (tree roots), cannot ensure the survival of their species. This observation was made during the study period when the most minimal fishing performance, both in terms of diversity and quantity of catches, were recorded in the villages of the LTCR, located in the North axis. Micha [28] also observed this fact in 2016. Hounsounou in 2011 [29] and 2013 [30] verified the same results in areas where gallery forests have been converted into shrubby savannah by bush fires, especially in Togo and Benin, including Epéna, Matoko Botakola, and Mokengui. This corroborates the analysis of the spatiotemporal dynamics, which revealed that the gallery forests were entirely converted into herbaceous, tree, shrub and slash, and burn savannahs between 1986, 1999, and 2019, to the detriment of the sustainability and durability of fishing activity, which is the main activity of the LTCR village communities.

The fisheries of the Likouala-aux-Herbes River, commonly called "spawning areas", were evaluated during the period from July to September 2019. A total of 204 spawning areas (i.e., 162 accessible areas and 42 prohibited areas) were identified in the seven pilot villages of the study in three LTCR fish basins. Overall, the ecological characteristics of these spawning grounds do not differ significantly from one station to another. The previous studies clearly show that the parameters of the Likouala-aux-Herbes River are

largely identical from upstream to downstream [31]. This further reinforces the results recorded during the study period in the various villages of the LTCR. The sampling stations were compared with each other, based on each hydromorphological parameter measured, independently of other factors, between July and September 2019. The variation in the width of the Likouala-aux-Herbes River changes from upstream to downstream. Thus, the Bouanela and Mossengue stations have larger beds. However, the depth did not significantly change from one station to another.

The Likouala-aux-Herbes River is a watercourse with a very low gradient due to the topography of the LTCR, [3], hence, the practically identical value ($3.43 \pm 0.73$ at Mokengui in the north of the LTCR and $3.30 \pm 0.78$ in the south at Bouanela) of the depth over the entire extent of the LTCR. On the other hand, width and depth are two hydromorphological parameters that contribute to the diversification of ichthyofauna in ichthyological hydrosystems. Several studies demonstrated relationships between species and environmental parameters in different hydrosystems. The distribution of Mormyridae in the Yoko River at Kisangani in the DRC was strongly correlated with a large steam width, as well as with depth [32]. According to Pwema [33] and Tomedi [34], the average water temperature in a rainforest stream rarely exceeds 24 °C. The average temperature recorded during the water sampling campaigns in the various pilot villages was mostly high in the LTCR, with values that fluctuated around 28 °C, with the exception of the Mokengui station, which recorded 25 °C. The similar value of 27.3 °C was obtained by Chikou [35] in the Bénin Ouémé Delta. The high riparian forest with shading caused by contiguous treetops that run along the river may be the source of this value, which supports the assertion of the impact of vegetation on temperature variation in the Congo Basin [36].

The Likouala-aux-Herbes River's waters are acidic, with pH values hovering around 4. The results obtained during the study are identical to those of Mamonekene [25], with values of $4.2 \pm 0.42$ into Mokengui and $4.35 \pm 0.13$ in Bouanela. The putrefaction of organic matter resulting from the continuous decomposition of drained plants (mudflat, wood debris, tree fruits, helophytes, and macrophytes) all along the river contributes through oxidation–reduction reactions to reduce the pH value of the river. The conductivity at 25 °C, recorded during the data collection campaign in the pilot villages of the LTCR, show higher values in the north of the reserve (90.40 μS/cm $\pm$ 27.65 in Mokengui and relatively low values in Bouanela in the south of the LTCR (58.20 μS/cm $\pm$ 24.50). These values are close to those of Mamonekene [25], whose conductivity values at 25 °C ranged from 56.5 μS/cm to 84.93 μS/cm.

The type of substrate encountered in the different stations of the Likouala-aux-Herbes River between July and September 2019 presents a great diversity from upstream to downstream and is mainly dominated by the mud–sand couple. Omasombo observed the same fact in Lac Mai-dombe in DCR [37]. A study carried out in Pool Malebo in the Congo River, for which the Likouala-aux-Herbes River, by way of the Sangha River, is one of the important tributaries, specifies that the diversity of substrates encountered in this part of the Congo River is the main explanation for the diversity of fish species present in the Pool Malebo fishery [38].

In addition to this diversity of station substrates, it is also important to mention that the diversity of vegetation, mainly encountered in the southern part of the reserve in the pilot villages of Mossengue, Botongo, and Bouenala also constitute fundamental elements to be taken into consideration in the description of the sampling stations. The aquatic vegetation in these villages is essentially made up of a variety of partially burned savannah formations, with a dominance of macrophytes helophytes, floating hydrophytes, *Eichornia crassipes*, fixed submerged *Echinochla* sp., and many others. In short, the physicochemical characteristics are identical in all the stations, particularly in the spawning grounds, showing that the diversity of the fish populations is more linked to the diversity of the microhabitats encountered in the Likouala-aux-Herbes River.

The interpretation of the data regarding the physicochemical and hydromorphological parameters, as well as the study of the substrate of each station (Table 5), showed that

the stations of the south axis of the LTCR have a greater width of area and a diversified aquatic vegetation that colonizes the banks and the bed of the watercourse, either in the Likouala-aux-Herbs River or floating on it.

All of this diversity of substrates in the waters of these localities greatly contributes to creating micro-habitats for fish species. The more that these microhabitats dependent on parameters, such as the width of the river bed and the depth, are diversified, the more the ichthyofauna of these localities will be diversified. The assertion that, in Ecoregion 539, tropical floodplain forest rivers consist of diverse micro-habitats, likely to be the source of high biodiversity and endemism was confirmed. The same observation regarding the distribution of *Labeo* (Cyprinidae) has been made in the Malebo Pool [38]. Their distribution is more related to the diversity of the micro-habitats encountered in the study area, whose main substrates were: mud, sand, stone, and silt.

In axis II of the PCA (Component 2, Figure 3), the distribution of the families of Alestidae and Mormyridae was influenced by the presence of sandy type substrates, foliage, and macrophytes, coupled with the increase in temperature. The presence of Alestidae is correlated with temperature peaks, namely 28.79 ± 1.7 °C for the Botongo station and 28.13 ± 1.17 °C for the Epena station (Tables 4 and 5). As for the Mormyridae, the highest diversity indices in this case, 0.6229 for Mossengue and 0.5993 for Botongo, were recorded at the stations with the highest temperature peaks. The families of Dichtichodontidae, Hepsetidae, Notopteridae, Anabantidae, Schlibeidae, and Polypteridae show no affinity either with the different substrates recorded or with the hydromorphological parameters of water axis III. However, they have affinities with each other, hence, their clustering in axis III. This affinity would be linked to the different trophic relationships (predator–prey) that exist between the ecological niches of these different species [38]. In short, the diversity of fish species encountered in the Likouala-aux-Herbs River results from our study of the different micro habitats existing inside and outside the bed of the watercourse.

## 5. Conclusions

In conclusion, the study on the contribution of remote sensing and GIS in the sustainable management of the fisheries resources of the Lac Télé Community Reserve on the Likouala-aux-Herbes River has made the following possible:

- To obtain primary knowledge of the dynamics of land use around the basin of the Likouala-aux-Herbes River in the LTCR. The LTCR is deteriorating under the probable effect of climate change, but also, and above all, due to human actions, which each year burn the floodplain and the gallery forest, degrading the potential spawning grounds;
- To obtain the distribution dynamics of fish species populations in the LTCR during the period from July to September 2019;
- To locate and characterize the 204 spawning grounds of the Likouala-aux-Herbes fisheries.

Regarding the spatiotemporal dynamics in the LTCR, the analysis of the land cover of the LTCR using Landsat MSS, ETM+, and OLI images revealed significant changes in the vegetation cover between 1986, 1999, and 2019. Thus, the forest cover area decreased by 21.41% for the entire LTCR, to the benefit of the savannahs, which increased in the same time periods. The ichthyological diversity recorded during the period from July to September 2019 in the seven pilot villages of the study of the three fishing basins of the Likouala-aux-Herbs River in the LTCR is due to the diversity of the micro-habitats noted in the villages of the axis south of the LTCR, as described in the characterization phase of the spawning grounds. Characterization of 151 spawning grounds by determining the hydromorphological parameters demonstrated that they did not significantly change between villages. However, we noticed a diversity of substrates in the spawning grounds of the villages of the southern axis of the LTCR, the main components of which were the mud–sand couple and the influx of aquatic vegetation composed of Helophytes, fixed and floating macrophytes on the waters of the Likouala-aux-Herbes River. Research opportunities that aim to strengthening the knowledge of biodiversity, in general, and ichthyology, in particular, in the LTCR in order to contribute to the preservation of this

RAMSAR site are a key asset for conservation, both in the Congo Basin, in general, and in the Republic of the Congo, in particular.

**Author Contributions:** Conceptualization, J.-C.M.; methodology, J.-C.M.; software, E.B.N.B.; validation, J.-C.M. and V.M.; formal analysis, E.B.N.B.; investigation, E.B.N.B.; resources, E.B.N.B.; data curation, E.B.N.B.; writing—original draft preparation, E.B.N.B.; writing—review and editing, J.-C.M.; visualization, E.B.N.B.; supervision, J.-C.M.; project administration, E.B.N.B.; funding acquisition, E.B.N.B. All authors have read and agreed to the published version of the manuscript.

**Funding:** Grant number ERAIFT ECOFAC 6-RCO/FED/039-795 and "The APC" was funded by the Wildlife Conservation Society (WCS) as part of the CARPE fund.

**Institutional Review Board Statement:** (ERAIFT) Regional Post-University School of Integrated Management of Tropical Forests and Lands (code 0034 and 9 November 2019).

**Data Availability Statement:** Data are contained within the article.

**Acknowledgments:** The authors thank the Wildlife Conservation Society (WCS) for the funding granted for the execution of the research work which led to these results. They also thank the University of Kinshasa (UNIKIN) and the Regional Post-University School of Forestry Planning and Management of Tropical Territories (ERAIFT) for their material and technical assistance in carrying out this research. The authors also extend their gratitude to the reference readers who were able to revise the manuscript for improvement. The authors thank the populations of the localities of the Likouala Department in the Republic of the Congo in general, and particularly those of the pilot villages that hosted the research work, for their hospitality and their generosity during the collection periods in these respective villages. Finally, thanks to Jennica Betsch who revised the English language of this manuscript.

**Conflicts of Interest:** The authors declare no conflicts of interest.

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
