# Peer review of "Characterization of Fish Spawning Grounds near the Likouala-Aux-Herbes River, Lac Tele Community Reserve (LTCR), Republic of Congo, for Sustainable Wetland Management"

_sustainability, doi:10.3390/su16083353_

Round 1
Reviewer 1 Report
Comments and Suggestions for Authors
1. Many of the French words and terminologies are kept in certain part of the article. However, it should be used consistently throughout.
2. The legend of land cover dynamic analysis in Figure A1 and the category of land cover in Table A1 and A2 are not the same. How do you correlate between Figure A1 and Table A1 and A2?
3. The components of PCA is not described as in Figure A2.
4. The fish were caught between July to September 2019. This short period of data does not have strength to conclude the fish diversity over land use change, degration of the fish productivity and the effect of land use change over biodiversity and water quality.
The quality is acceptable.
Author Response
Answers of questions
- Many of the French words and terminologies are kept in certain part of the article. However, it should be used consistently throughout.
It's ok. ALL the french word and terminologies are used correctly int the article
2. The legend of land cover dynamic analysis in Figure A1 and the category of land cover in Table A1 and A2 are not the same. How do you correlate between Figure A1 and Table A1 and A2?
It's Ok , we have the same category to compare in Table A1 and A2
3. The components of PCA is not described as in Figure A2
It 's ok, All the components of PCA are described in Figures A2, table 4 and Table 5
4. The fish were caught between July to September 2019. This short period of data does not have strength to conclude the fish diversity over land use change, degration of the fish productivity and the effect of land use change over biodiversity and water quality.
It's true , that is reason I precise in abstract and discussion " can be'' the cause of degradation of fisheries biodiviersity. We need to complete this study with others data to conclude that the degradation of forest to savanah decrease the stock fisheries.
Reviewer 2 Report
Comments and Suggestions for Authors
1- The abbreviation of Lac Télé Community Reserve (RCLT) is wrong and should be as LTCR. It should be corrected in all parts.
2- English polish is recommended.
3- Highly recommended to insert a study area figure, to show where your study area is?
I mean you should present the location of your area in your county. Like other manuscripts.
4- How did the land cover maps were generated? As example:
You should explain the train and test samples for classification
Classification algorithm(s)
Classification accuracy assessment results (as confusion matrix)
And all relevant contents should be inserted in your manuscript.
Author Response
- The abbreviation of Lac Télé Community Reserve (RCLT) is wrong and should be as LTCR. It should be corrected in all parts.
It’s Ok, RCLT changes to LTCR in the article.
- English polish is recommended.
Some sentences have been ameliorated in all the parts of article
3- Highly recommended to insert a study area figure, to show where your study area is?
I mean you should present the location of your area in your county. Like other manuscripts.
It’s ok. Figure A1 ; location of study’s area in Republic of Congo
4- How did the land cover maps were generated? As example:
You should explain the train and test samples for classification
Classification algorithm(s)
Classification accuracy assessment results (as confusion matrix)
And all relevant contents should be inserted in your manuscript.
It ‘s ok. The methodology of remote sensing was explained in Materiels and methods.
Reviewer 3 Report
Comments and Suggestions for Authors
The methodology is poorly described. The name of the paper includes the main key word: Remote sensing. I don't see anything about remote sensing methods used nor how it corresponds and is connected to the research. If you just remove the word Remote sensing from the name of the paper then the whole paper will go into a different thematic group of water quality and fisheries....

Some of the sentences are not understandable, and the wording should be revised.
Author Response
The methodology is poorly described. The name of the paper includes the main key word: Remote sensing. I don't see anything about remote sensing methods used nor how it corresponds and is connected to the research. If you just remove the word Remote sensing from the name of the paper then the whole paper will go into a different thematic group of water quality and fisheries....
It ‘s ok. The methodology of remote sensing was explained in Materiels and methods.
Reviewer 4 Report
Comments and Suggestions for Authors
In this study, remote sensing images are utilized to examine how wetlands are managed for sustainable development, including changes in land use. Although the paper has some practical use, there are numerous issues. The particular issues are as follows:
1. "LTCR" in the latter, "RCLT" in the Abstract;
2. "07" need to read "7";
3.Line 27, please check to see if the coordinate position representation technique is correct;
4. "Through"'s first letter needs to be capitalized. The complete text should be checked and revised because there are other issues of this paper.
5.Line 37's initial citation should begin with [1] rather than [6], and the complete text should be checked and modified;
6. Section 2, which describes the paper's methodology, needs to be enhanced and supplemented; "NON-PUBLISHED MATERIELS" need to serve as an introduction to the methodology; why not include it in the main text?
7.Lines 68, 82, and 83's incorrect unit format for speed and area should be checked and the entire text revised;
8. Why is the letter "A" included to figures and tables?
9. This work primarily examines the use of remote sensing and GIS in the management of fishery resources, although it is advised that it be supplemented with more detailed descriptions of these applications.
10.Line 423–466 should be modified in accordance with the journal format guidelines;
11. There aren't enough references, particularly in recent years.
Author Response
In this study, remote sensing images are utilized to examine how wetlands are managed for sustainable development, including changes in land use. Although the paper has some practical use, there are numerous issues. The particular issues are as follows:
- "LTCR" in the latter, "RCLT" in the Abstract ;
It ‘s ok, I correct it.
- "07" need to read "7";
It ‘s ok, I correct it.
3.Line 27, please check to see if the coordinate position representation technique is correct;
It ‘s ok, It change it
- "Through"'s first letter needs to be capitalized. The complete text should be checked and revised because there are other issues of this paper.
- Some sentences have been ameliorated in all the parts of article
5.Line 37's initial citation should begin with [1] rather than [6], and the complete text should be checked and modified;
It’s ok. The citation started with [1]
- Section 2, which describes the paper's methodology, needs to be enhanced and supplemented; "NON-PUBLISHED MATERIELS" need to serve as an introduction to the methodology; why not include it in the main text?
It’s ok. Tilte of section 2 is : NON-PUBLISHED MATERIELS
7.Lines 68, 82, and 83's incorrect unit format for speed and area should be checked and the entire text revised;
It’s ok ; the paragraph is correct.
The sampling unit was a pilot village in LTCR. the three values have been collected around the site to get a reference value for the physic and chemical parameters as; deep (m), width (m ); velocity (m/s); debit (m3/s); transparency (m); color of water; temperature (°C); pH; conductivity which were collected at 25°C.
Those data were collected every month in the same period according to the Zebaze [4] who confirm and conclude that this frequency accounts for the functioning of hydrosystems. Physic and chemical water parameters; temperature, pH and conductivity at 25°C were measured by using the HACH IntelliCAL phC101 multi-parameter (b). Water depth (c) and transparency (d) were measured by using a water level gauge and Secchi disk, respectively.
The width of the sampled section was measured by using a decameter. The water velocity was measured in m/s, by using an empirical method which consisted by recording the distance and the time traveled a stopwatch by a float immersed and followed over a certain distance. Subsequently the debit was deducted from the formula.
Q = V X S with V = velocity in m/s
S = area in m2
Q = debit in m3/s
- Why is the letter "A" included to figures and tables ?
Writting methodology of MPDI recommended to put letter A before numerous of table and figure.
- This work primarily examines the use of remote sensing and GIS in the management of fishery resources, although it is advised that it be supplemented with more detailed descriptions of these applications.
It ‘s ok. The methodology of remote sensing and using GIS was explained in Materiels and methods.
10.Line 423–466 should be modified in accordance with the journal format guidelines ;
It ‘s ok. The references were modified in accordance.
- There aren't enough references, particularly in recent years.
I didn’t found others articles wtih the same results which can help us to discuss our result.
Round 2
Reviewer 2 Report
Comments and Suggestions for Authors
Thanks for making the requested corrections and good luck in other scientific work.
Author Response
Comment and suggestions for authors are ok.
Reviewer 3 Report
Comments and Suggestions for Authors
This time there is some material for Remote sensing. The remote sensing method is still not very clear. Citation: The land cover maps were the result of the interpretation of satellite images. I understand this as manual photo interpretation. So this means manual digitalization of land cover polygons. But from the maps I see small pixels, and probably this was done with semiautomatic pixel based classification. Also there are a lot of French words in the text legends. This should be changed into English.
Comments on the Quality of English LanguagePlease check the English, there are a lot of mistakes.
Author Response
For Comments and suggestions
According to classification. Yes semiautomatic classification. I precise it in the methodology.
I also present the translation of french words in the table 1 and table 2. It's not possible to change the maps now.
Reviewer 4 Report
Comments and Suggestions for Authors
1. Lines 11 and 15, "LTCR" in the main text, "RCLT" in the Abstract; the issue is still present.
2. "NON-PUBLISHED MATERIELS" ought to serve as the method's introduction; instead, include it in the main text. No reasonable explanation was given.
3.106 line should be Conversive, 111 line is Changes.
4. The use of GIS and remote sensing in fishery resource management is still insufficient since it only covers the data processing flow and source description.
5. References can search for related research areas, albeit they don't have to be the same.
Author Response
- Lines 11 and 15, "LTCR" in the main text, "RCLT" in the Abstract; the issue is still present. answer: It is ok
- 2. "NON-PUBLISHED MATERIELS" ought to serve as the method's introduction; instead, include it in the main text. No reasonable explanation was given/ answer: I present all the materiels in the methodology which have been used during the study. I dont need to present the pictures of those materiels
- 106 line should be Conversive, 111 line is Changes. answer: It's ok
4. The use of GIS and remote sensing in fishery resource management is still insufficient since it only covers the data processing flow and source description.
We also present the use of GIS and remote sensing to contribute to the compare fisheries at different times.
5. References can search for related research areas, albeit they don't have to be the same
Thanks for this suggestion
Round 3
Reviewer 4 Report
Comments and Suggestions for Authors
The document also contains a few small mistakes, such as a formula unit format error in lines 85 and 86.
Author Response
The document also contains a few small mistakes, such as a formula unit format error in lines 85 and 86.
Answer : IT's ok. I correct the mistake